# Neural Differences in Relation to Risk Preferences during Reward Processing: An Event-Related Potential Study

**DOI:** 10.3390/brainsci13091235

**Published:** 2023-08-24

**Authors:** Sedigheh Naghel, Antonino Vallesi, Hassan Sabouri Moghadam, Mohammad Ali Nazari

**Affiliations:** 1Department of Neuroscience, Faculty of Psychology and Educational Science, University of Tabriz, Tabriz 5166616471, Iran; nagel_s@tabrizu.ac.ir (S.N.); sabouri-h@tabrizu.ac.ir (H.S.M.); 2Department of Neuroscience & Padova Neuroscience Center, University of Padova, 35128 Padova, Italy; 3Department of Neuroscience, Faculty of Advanced Technologies in Medicine, Iran University of Medical Sciences, Tehran 1449614535, Iran

**Keywords:** risk preferences, reward processing, ERP, reward anticipation, reward outcome

## Abstract

Inter-individual variability in risk preferences can be reflected in reward processing differences, making people risk-seekers or risk-averse. However, the neural correlates of reward processing in individuals with risk preferences remain unknown. Consequently, this event-related potential (ERP) study examined and compared electrophysiological correlates associated with different stages of reward processing in risk-seeking and risk-averse groups. Individuals scoring in the bottom and top 20% on the Balloon Analogue Risk Task (BART) were deemed risk-averse and risk-seeking, respectively. Participants engaged in a gambling task while their electroencephalogram (EEG) was recorded. Risk-seekers tended to choose high-risk options significantly more frequently than low-risk options, whereas risk-averse individuals chose low-risk options significantly more frequently than high-risk ones. All participants selected the low-risk alternative more slowly than the high-risk option. During the anticipation stage, the low-risk option elicited a relatively attenuated stimulus-preceding negativity (SPN) response from risk-seekers compared to risk-averse participants. During the outcome stage, feedback-related negativity (FRN) increased in risk-seekers responding to greater losses but not in risk-averse participants. These results indicate that ERP components can detect differences in reward processing during risky situations. In addition, these results suggest that motivation and cognitive control, along with their associated neural processes, may play a central role in differences in reward-based behavior between the two groups.

## 1. Introduction

In daily life, the extent of willingness to take risks is referred to as risk preference [1], a concept that ranges from risk-aversion to risk-seeking, with individuals on the two extremes of this continuum preferably selecting low-risk and high-risk options, respectively [2]. Risk preference has generally been regarded as one of the most important constructs in psychology and behavioral science [3], particularly as a crucial index of risky decision-making [4]. Increasing evidence suggests that risky decision-making is associated with reward-based differences in the midbrain dopaminergic system, such as the ventral striatum and frontal areas, including the ventromedial prefrontal cortex [5,6]. Various studies have demonstrated that abnormalities in these reward-processing regions are associated with maladaptive risk-taking in clinical populations [6], including alcohol use disorder [7], gambling behaviors [8], internet gaming disorder [9], and cannabis users [10].

According to the “reward deficiency syndrome” hypothesis, people with high-risk preferences (such as impulsive and addictive behaviors) have a deficiency of dopaminergic receptors [11], resulting in poor reward processing [12]. Therefore, changes in the activity of the dopaminergic system could play an essential role in modulating risk preferences and make people more risk-averse or risk-seeking during reward processing [12,13,14]. Moreover, changes in the dopaminergic system modulate the activity of the anterior cingulate cortex (ACC), which in turn influences top-down cognitive control processes [15]. Based on the error-likelihood model [16], the activity of ACC represents risk perception and differences in risk aversion. This model assumes that the activity of ACC concerning risk prediction and reward magnitude detects responses to conflict and facilitates control processes in risk-averse individuals. In contrast, risk-seeking individuals show higher ACC activity due to a higher level of response conflict and a lower level of cognitive control [16].

ACC and dopaminergic activity are not the only sign of risk preferences in reward processing. Studies have shown that harm-avoidant subjects exhibit greater activation in the right insular cortex [17] while choosing to avoid punishment during the anticipation phase of reward processing [18,19]. According to the somatic marker hypothesis [20], aversive stimuli are associated with somatic markers, leading to stronger activation of the insula [21]. This relatively strong activation indicates the possibility of aversive feedback and prompts the subject to avoid risky options [17]. Consistent with this view, some studies showed that decreased insula activity was correlated with craving for alcohol [22] and pathological gambling [19].

As neuroimaging evidence shows, the ACC and the insula play a crucial role in determining risky decisions during reward processing. Classically, reward processing is divided into two temporally distinct phases: reward-anticipation and reward-outcome processing [23,24,25,26], which involve internal motivational tendencies related to approaching reward stimuli and pleasant emotions due to reward attainment, respectively [9,25,26,27]. The history of research on reward processing shows that these phases are temporally and anatomically distinct [26], then selecting a recording method with high temporal resolution is key to studying reward processing behaviors. In general, neuroimaging techniques do not have the temporal resolution to distinguish all cognitive and neural activities underlying these phases [6]; therefore, ERP recording was used in this study due to its high temporal resolution, in the millisecond range [28].

In this context, researchers have primarily focused on feedback-related negativity (FRN), a negative-going ERP component that occurs 200 to 350 ms after feedback is received [26,29,30]. The FRN is considered an error signal for reward prediction [28] that is sensitive to valence and magnitude [31]. The FRN is more pronounced for negative feedback than positive feedback and peaks at frontocentral electrodes [32,33] during the reward-outcome stage. Some studies have shown that the phasic activity of dopamine neurons in the ACC modulates the amplitude of the FRN [34,35]. There is also evidence of a positive relationship between FRN amplitude and risk-averse traits [36].

Another visible component before feedback presentation is the so-called stimulus-preceding negativity (SPN), which correlates with passive reward anticipation and is measured about 200 ms before feedback presentation [26]. This slow cortical potential has been shown to correlate with motivational processes during the anticipation stage [37,38,39] and to have an amplitude maximum in right frontocentral areas [26,38,40] and to be sensitive to reward probability and motivational salience [41]. The insula has been shown to be the primary generator of the SPN and its activation modulates SPN amplitude [39,42]. Therefore, it is reasonable to assume that differences in risk preferences might also be reflected in SPN amplitude. As the present study targeted reward processing and risk preferences, FRN and SPN were selected based on their association with risk and reward-related regions, including ACC and insula, and their temporal correlation with stages of reward processing, including anticipation and feedback. The literature has also focused on the P300 component, due to its correlation with the reward-outcome phase. This component has been discussed in more detail in another article by these authors in Persian [43].

Prior research focused primarily on risky behaviors during reward processing in clinical populations, such as those with alcohol use disorders [7], gambling behaviors [8], internet gaming disorders [9], and cannabis users [10]. However, non-clinical populations with different risk preferences in economic conditions have been the subject of few studies. In one study, Polezzi and colleagues [44] examined the neural basis of risky behavior in various contexts. They concentrated primarily on outcome processing and observed that risk-taking is primarily reflected by early evaluation of the outcome, as represented by feedback-related negativity (FRN). Other studies have focused on various personality traits associated with risky behavior, including impulsivity [31] and sensation-seeking [45], and examined their event-related potential (ERP) differences in reward stages. To the best of our knowledge, no electroencephalogram (EEG) study has examined the differences in ERP components between risk-seeking and risk-averse healthy individuals during the anticipation and feedback stages of reward processing.

The current study aimed to compare the stages of reward processing between two non-clinical groups with different risk preferences. To this end, the Balloon Analogue Risk Task (BART) categorized healthy participants as risk-seekers and risk-averse. It has been shown that BART performance is associated with real-world risk-taking, smoking, gambling, and substance abuse [31]. During the EEG recording, participants chose between low-risk and high-risk options in a simple gambling task. It was hypothesized that risk preferences would be distinguished by differences in stimulus-preceding negativity (SPN) during the anticipation stage and FRN during the outcome stage. In light of the reward deficiency syndrome theory, it can be hypothesized that risk-seekers would exhibit a smaller FRN amplitude in response to negative feedback or losses than risk-averse individuals.

In contrast, based on the error-likelihood model and reward salience hypothesis, it can be predicted that, when the reward is salient and valuable in magnitude, risk-seekers would exhibit greater central scalp activity possibly indicating anterior cingulate cortex ACC activation in response to losses. Conversely, according to the somatic marker hypothesis and insular activation, risk-averse individuals are predicted to exhibit greater SPN during reward anticipation. Finally, this study investigated the potential relationship between reward anticipation as indexed by SPN and reward outcome as indexed by FRN, which could provide insight into neural correlates of risky decision-making dynamics during reward processing.

## 2. Materials and Methods

### 2.1. Participants

Thirty-one participants were selected from 352 undergraduates based on their scores in the Persian version of Balloon Analogue Risk Task (BART) [45]. An adjusted score, the average of pumps in unexploded balloons, was used to measure risk-taking [46]. Higher scores on the adjusted value indicate higher risk-seeking and vice versa. The risk-seeking and risk-averse groups were assigned according to scores in the top 20% (range = 55–75) and bottom 20% (range = 8–19) of the distribution of risk-taking scores (M = 37.71, range = 8.44–85), respectively. Selected participants (16 risk-seekers and 15 risk-averse) were invited to perform an EEG examination. The aims of the study were explained in a brief statement indicating participation in a study of cognitive and brain evaluation using computerized tasks and brain recording voluntarily, and participants signed a consent form before starting the study. Twenty-seven participants were right-handed and four left-handed (1 risk-averse and 3 risk-seekers). No participant reported a history of neurological disorders, psychiatric disorders, or, more specifically, substance dependence.

### 2.2. The Balloon Analog Risk Task (BART)

Using the Balloon Analog Risk Task subjects were divided into risk-seekers and risk-averse. The BART is a computer-simulated test that measures the construct of impulsivity through a risk-taking attitude [47]. This task classically involves earning virtual money by inflating a virtual balloon after pressing a pump button. In the present study, each time the pump button was pressed the balloon inflated by 1֯ (about 0.3 cm) in all directions, and 500 Iranian Rials (IRR) were reserved or deposited into a temporary virtual bank. Participants could either collect the money by pressing the collection bottom and transferring the money from the temporary to the permanent bank or lose the money when the balloon burst with an explosion sound (as a result of randomly reaching individual explosion points of each balloon). A new, uninflated balloon appeared either after the collection was pressed or after the balloon exploded (30 balloons in total). Adjusted values were calculated for all participants, defining the average of balloon pumps in balloons that did not explode [46].

### 2.3. Task and Procedure

During the EEG recording, participants sat in front of a computer screen on which they presented a simple gambling task. The task was derived from the gambling task of Gehring and Willoughby’s [48] study. In this task, participants choose between two amounts of monetary (here IRR 100 and 500) and receive random feedback on whether they won or lost the chosen amount. The target stimuli consisted of two squares containing the numbers (100 and 500), displayed for 3000 ms. Feedback stimuli were presented after a 2000 ms interval of a blank screen. They resembled the target cards in shape and size, but their color changed to green or red, indicating a win or a loss, respectively. The color of the unchosen card also changed to indicate what the outcome would have been if the subject had made the other choice. Thus, the results included four main conditions: lose and correct, gain and correct, lose and error, and gain and error. For example, a win of 100, when the unchosen card is the win of IRR 500 associated with the gain and error condition, suggests that choosing 500 would have meant more gain. All possible conditions are displayed in Table 1.

The task was presented by eevoke™ software (ANT neuro, b.v., Hengelo, Netherlands). Each trial started with a fixation point with a 500 ms duration and 500 ms inter-trial interval (ITI). The first stimulus was displayed for 3000 ms including choices in gray squares and a white background. Then, a white blank page was presented for 2000 ms as the inter-stimulus interval (ISI). The feedback stimulus was displayed for 2000 ms. The task is detailed in Figure 1. Participants were instructed to choose rapidly as the first stimulus was displayed and were unaware of the random presentation of feedback stimuli. The total number of trials was 240. In order to gain accurate and correct results, there was a reward according to the participant’s scores at the end of the recordings, and a large reward for the subject who collected the most money (USD 10 in Iran’s currency).

### 2.4. EEG Recording and Data Processing

The EEG was recorded using an ANT neuro amplifier with ASA-lab recording software (product version 4.7.1, ANT neuro, b.v., Hengelo, Netherlands). It was recorded through the elastic Waveguard^TM^ cap with a set of 64 shielded Ag/AgCl electrodes according to the international 10–10 system attached to a 64-channel EEG/ERP ASA-Lab system by shielded cables. The EEG signals were referenced online to the left and right mastoid electrodes and filtered between 0.5 and 50 Hz, at a sampling rate of 256 Hz. Impedance was held <10 KΩ for all electrodes.

The EEG data were preprocessed and analyzed using MATLAB 2015a (Math Works INC, Natick, Massachusetts) and EEGLAB toolbox [48]. The original EEG signals were filtered twice using a different parameter with a low-pass filter set at 30 Hz for SPN analysis and a band-pass of 0.5 to 30 Hz for FRN. Eye-blink artifacts were identified and corrected by independent component analysis (ICA).

Filtered EEG data were fragmented into epochs which were time-locked to the feedback onset using the ERPLAb toolbox (v 6.1.4) [49]. The epoched data for each participant were screened manually for artifacts (e.g., spikes, and non-biological signals). In order to analyze SPN, epochs began 1000 ms prior to the feedback onset and ended 500 ms after that, and the baseline was from −1000 to −800 ms [44,50]. Then, SPN was measured as the mean amplitude from the 200 ms directly before feedback onset (namely a window between −200 to 0 ms). It was maximal at right central electrode sites such as C2. For the FRN, epochs included 200 ms prior to the feedback activity and extended 800 ms post-feedback, with the activity from −200 to 0 ms serving as the baseline. In order to measure FRN, the peak amplitude from 200 to 350 after the feedback onset was extracted. It was more pronounced at FZ, C2, and Cz electrodes, consistent with the previous literature [51,52,53].

Behavioral data analysis was performed on two dependent variables: choice percentage and selection time of choice. The choice percentage corresponded to the selection frequency of 100 as the low-risk option and 500 as the high-risk option. The selection time of choice was measured by the mean time interval from the onset of the stimulus to the selection of an option. Two separate mixed Analyses of Variance (ANOVAs) were used for each dependent variable, separately. In this regard, risk preferences (risk-seeker vs. risk-averse) were considered as the between-subjects factor, and choice (high risk vs. low risk) was the within-subject factor.

In order to analyze ERP data, the mean amplitude of SPN and peak amplitude of FRN components were analyzed. The SPN data were analyzed using group (risk-seeker vs. risk-averse) as a between-subjects factor and choice (low-risk ‘100′ vs. high-risk ‘500′) as the within-subject factor. The FRN data were analyzed with a group × feedback mixed ANOVA, including group (risk-seeker vs. risk-averse) as a between-subject factor and feedback condition (4 levels: low loss ‘−100′, large loss ‘−500′, low gain ‘+100′, large gain ‘+500′) as within-subject factor. Statistical analyses were conducted using SPSS (Version 19.0, Armonk, NY: IBM Corp) software.

## 3. Results

### 3.1. Demographic and Behavioral Data

Table 2 presents the behavioral data for risk-seeker and risk-averse groups, respectively. Groups showed no difference in age and gender (ps > 0.05), and their adjusted number on BART was significantly different (ps < 0.05), confirming the difference in risk-preference level between groups. Barratt’s impulsiveness scale (BIS) results were also analyzed based on an anonymous reviewer’s suggestion.

Results showed that the main effect of choice was significant, representing those participants choose the high-risk option (500) more than the low-risk option (100) [F(1,26) = 22.64, *p* < 0.001, η_p_^2^ = 0.466]. In addition, the interaction effect of group and choice was significant [F(1,26) = 5.17, *p* = 0.031, η_p_^2^ = 0.166], showing the difference between groups in selecting options (Figure 2A). In order to further explore the difference between groups in each choice, an independent *t*-test analysis was used. The results showed that the risk-averse group significantly chose more low-risk compared to risk-seekers [*p* = 0.041]. In contrast, another independent *t*-test revealed that risk-seekers significantly had more selection of high-risk options compared to risk-averse [*p* = 0.049].

The data showed that the main effect of selection time was significant and the comparison of the mean indicated that subjects chose 100 (low-risk option) more slowly than 500 (high-risk option) [F(1,26) = 21.81, *p* < 0.001, η_p_^2^ = 0.456]. However, the interaction between group × selection time was not significant [F(1,26) = 2.67, *p* = 0.229, η_p_^2^ = 0.055].

### 3.2. Electroencephalography Data

#### 3.2.1. SPN

Figure 3 displays the grand-average ERP waveforms preceding feedback presentation and topographic maps of the SPN in the interval from −200 to 0 ms. Consistent with the study by Brunia and colleagues [38], SPN was larger at central and right electrodes and reached maximum prior to the feedback onset. Although the SPN was larger following the high-risk choice (500) compared to the low-risk choice (100), the main effect of choice was not significant [F(1,26) = 0.935, *p* = 0.343, η_p_^2^ = 0.035], indicating that, regardless of groups, choice risk had no effect on SPN amplitude.

Furthermore, there was a significant interaction effect of the group by choice [F(1,26) = 4.412, *p* = 0.046, η_p_^2^ = 0.145], representing the influence of risk preferences on the SPN amplitude.

Further analyses using the independent *t*-test were performed to determine the group differences in conditions. It revealed that there was a significant difference in SPN amplitude between groups after selecting the low-risk choice [t = 3.056, *p* = 0.005], such that the risk-averse group displayed a larger SPN amplitude after selecting the low-risk choice (Figure 4).

#### 3.2.2. FRN

Figure 5 depicts the grand-average ERP waveforms and topographic maps of FRN elicited by feedback between the two groups. FRN was measurable as a frontal negative ERP component peaking at approximately 200 to 350 ms after feedback presentation. Feedback including losses elicited a more pronounced FRN than gains did, as reflected by the significant main effect of feedback [F(1,26) = 4.58, *p* = 0.005, η_p_^2^ = 0.150].

A further analysis displayed a significant interaction between groups and feedback [F(1,26) = 2.98, *p* = 0.036, η_p_^2^ = 0.103]. This effect was qualified by a significantly larger FRN after the large loss (−500) in risk-seekers compared to the risk-averse group as shown by an exploratory independent *t*-test (Figure 6) [t = 7.02, *p* = 0.007].

## 4. Discussion

The main objective of this study was to examine the neural differences of risk preferences in neural correlates of reward processing using risk-specific ERP components (i.e., the FRN and SPN) while the participants performed a gambling task. While all participants had a significantly greater tendency to choose the high-risk option, the behavioral analysis revealed that the risk-averse group chose low-risk options significantly more often than the risk-seeking group. On the other hand, risk-seekers chose the high-risk option significantly more frequently than the risk-averse group. This result is consistent with typical risk-seeking and risk aversion in individuals, as demonstrated by previous studies [45,54]. Risk-averse individuals have a strong propensity to select the option with the lowest risk potential [55]. Conversely, risk-seekers tend to select options with high-risk potentials that may result in substantial loss. In addition, regardless of risk preferences, all subjects chose high-risk options faster than low-risk options.

This section looks at risk preference neural differences and the relevant EEG results. According to ERP analysis, risk-averse individuals exhibited greater SPN amplitude in all conditions, irrespective of the magnitude of the choice. In contrast, risk-seekers demonstrated a diminished response to the safe option. This result confirmed our prediction and concurred with the findings of previous studies [17,45]. Zheng and Liu [45] observed that individuals with an elevated level of sensation-seeking have a blunted SPN compared to those with a low level of sensation-seeking. As previously stated, the insula is recognized as the primary generator of the SPN component [56]. Following the somatic marker hypothesis, the insular cortex mediates between the internal physiological state and external cues. Paulus and colleagues [17] proposed that the higher activation of the insula represents aversive somatic markers due to the perception of potential aversive outcomes. In addition, they found a positive correlation between insular activation and the degree of harm avoidance. The current study revealed that this correlation is measurable by the amplitude of the SPN component during the anticipation stage.

The results reconfirmed that SPN is associated with affective anticipation [51] and is increased by uncertainty and threat [57]. Since fear of uncertainty implies a significant sense of anxiety [58] and previous studies showed a positive correlation between risk-averse traits and anxiety levels [59,60,61], the more negative SPN in risk-averse individuals can be speculatively interpreted as higher levels of perceived uncertainty and anxiety. The results are also supported by imaging studies examining the role of the insula during anticipation [17,56,62,63], particularly with respect to its role in signaling and triggering avoidant behavior [56]. According to the work of Alvarez and colleagues [63], this aversive bodily state causes anticipatory anxiety due to the activation of the anterior insula, which primarily reduces perceived control. Therefore, we might speculate that risk-averse people as anxiety-prone subjects showed a higher level of insula activation represented by SPN because of their decreased perceived control. Based on the present results, there was no significant difference in SPN amplitude following safety and risky choice across all subjects, which is inconsistent with the results of previous research [45,64,65].

In line with previous research, the FRN amplitude was significantly greater in response to losses than gains in both groups during the feedback processing stage [31,34,66]. This result demonstrates the correlation between the FRN component and response conflict, as established by previous research [45,67,68]. It is hypothesized that the FRN component represents the emotional evaluation of the feedback [36], especially when the feedback is worse than expected [69]. This binary evaluation of good versus bad [70] is modulated by the dopaminergic system during the feedback stage [36] and can be indexed by changes in FRN amplitude.

Regarding the extent of negative feedback, risk-seekers showed increased FRN in response to greater loss (500). There is a contradiction in the previous literature on risky behaviors and FRN. For example, in Zheng and Liu’s study [45], FRN was more negative for individuals with low sensation-seeking after high-risk options. This effect was interpreted as higher motivational salience in this group. In contrast, in the study by Teti Mayer and colleagues [31], there was no difference in FRN amplitude between impulsive and healthy individuals, while current subjects showed no significant differences in the BIS scale (see Table 2). Therefore, these discrepancies could be due to differences in the measurements used to classify subjects into groups. In this regard, future research needs to examine and compare ERPs between personality traits correlated with risky behavior using different measures.

The increased FRN in the risk-seeker group is consistent with the error-likelihood model in the study by Brown and Braver [16]. According to this model, ACC plays an essential role in measuring expected risk [71] and conflict processing [72], especially as this role is mediated by higher levels of cognitive control and error-likelihood effects. Thus, when risk avoiders choose a riskier option, they predict more loss or error likelihood which increases their cognitive control. This increased cognitive control reduces conflict effects and leads to lower ACC activity [17]. In contrast, risk-seekers showed higher ACC activity, which is reflected by the FRN, because of attenuated cognitive control and less error prediction. Other fMRI studies also support this hyperactivity of reward regions in risk-takers. For example, Yarosh and colleagues [5] found greater ventral striatal activation in subjects with a family history of alcoholism, interpreted as a pro-motivational drive and reduced cognitive control [5].

It can be concluded that risk-seekers may show greater ACC activity due to deficits in cognitive control when the possibility of loss is high. This is also consistent with the reward salience hypothesis, which states that reward saliency about cognitive control leads to the over-activation of ACC in substance users [73]. As mentioned earlier, according to the reinforcement theory of FRN in the study by Holroyd and Coles [34], this component reflects the extent of loss compared to what was expected. The result of the present study shows that this reward prediction error is more pronounced in the individual with higher risk-seeking levels. This finding is inconsistent with the prediction of the reward deficiency syndrome hypothesis, which predicts hypo-activity in the regions of the reward system in high-risk individuals.

Our study was not free of limitations. One limitation concerns the small sample size which might raise power and replicability issues. Thereupon, future research should include more subjects to obtain larger and more robust effects. Another limitation stems from the fact that different psychological measurements relating to risk and more demographic and situational information (e.g., menstruation in women, socioeconomic situation) were not collected and their role could not be analyzed.

Overall, this study showed that the ERP components can be a fundamental tool for comparing and determining inter-individual differences in the reward processing stages. Moreover, the result suggests that the pre-reward expectations and post-reward analysis may be influenced by risk preferences.

## Figures and Tables

**Figure 1 brainsci-13-01235-f001:**
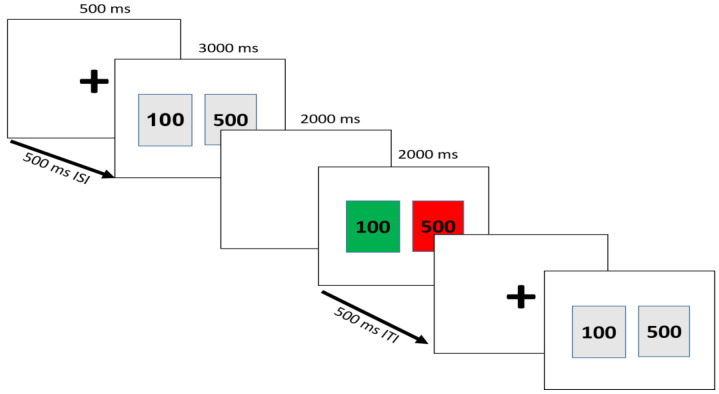
Sequence of possible events and stimulus types in the gambling task.

**Figure 2 brainsci-13-01235-f002:**
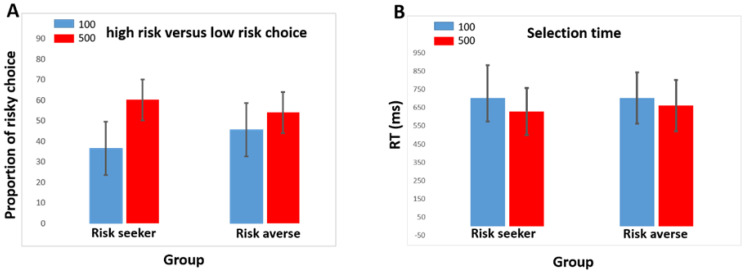
Behavioral data: (**A**) stimulus selection; highrisk versus low-risk choices at risk-seeker and risk-averse group, (**B**) selection time or decision-making time to select each choice for risk-seeker and risk-averse groups.

**Figure 3 brainsci-13-01235-f003:**
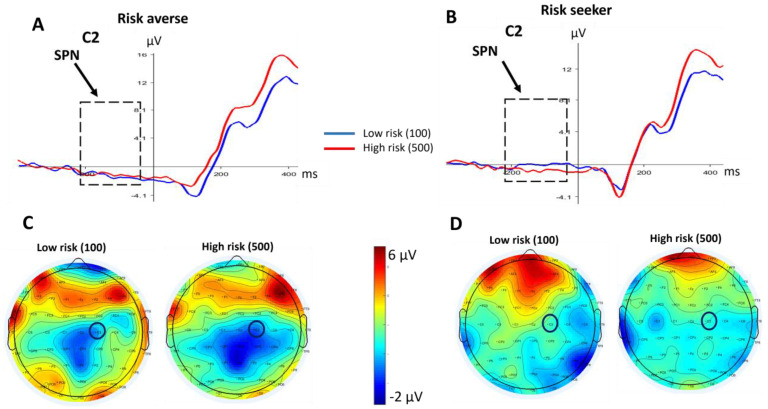
(**A**) Grand-average ERP waveform of the risk-averse group following choices at C2. SPN is identified with the dashed line rectangle. (**B**) Grand-average ERP waveform of risk-seeker group following choices. (**C**) Scalp maps (−200–0 ms) show the topography of the SPN in the risk-averse group. The C2 electrode has been highlighted with a black circle. (**D**) Scalp maps (−200–0 ms) show the topography of the SPN in the risk-seeker group.

**Figure 4 brainsci-13-01235-f004:**
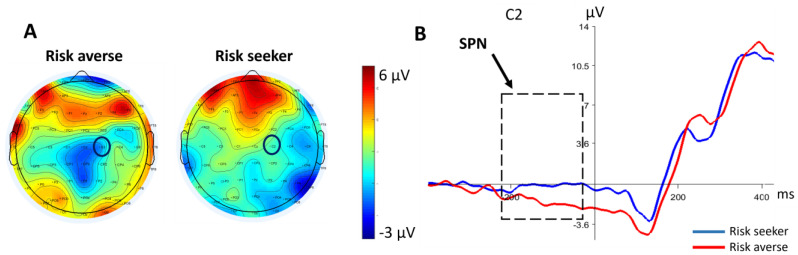
(**A**) Scalp maps of both groups, showing topography for the SPN (−200–0 ms) following low-risk choice. The C2 electrode has been highlighted with a black circle. (**B**) Comparing grand-average ERP waveform between groups following low-risk choice (100) at C2. SPN has been identified with the dashed line box.

**Figure 5 brainsci-13-01235-f005:**
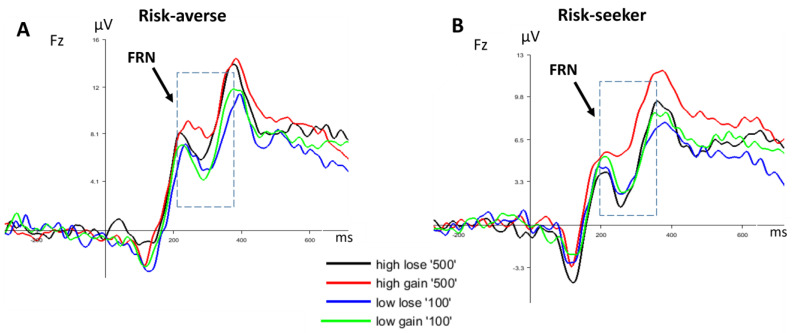
(**A**) Grand-average ERP waveform of the risk-averse group after receiving four feedbacks (low gain and high gain/low loss and high loss) at Fz. FRN has been identified with the dashed line box. (**B**) Grand-average ERP waveform of risk-seeker group after receiving four feedbacks. (**C**) Scalp maps (250–350 ms) show the topography for the FRN at the risk-averse group after receiving feedbacks. The Fz electrode has been highlighted with a circle. (**D**) Scalp maps (250–350 ms) show the topography for the FRN at the risk-seeker group after receiving feedbacks.

**Figure 6 brainsci-13-01235-f006:**
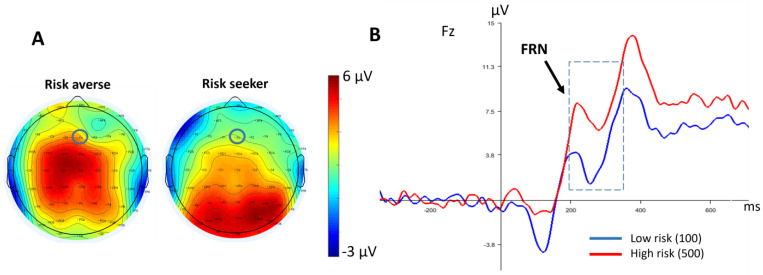
(**A**) Scalp maps of both groups show the topography of FRN. The Fz electrode has been highlighted with a black circle. (**B**) Comparing grand-average ERP waveform between groups following the high loss (−500) feedback at Fz. FRN has been identified with the dashed line box.

**Table 1 brainsci-13-01235-t001:** Possible conditions according to the selected card in the simple gambling task. The red and green colors describe loss and gain in the gambling task.

Feedback Target		Chosen Card	Condition
100	500	100	+100 Gain and correct
		500	−500 Lose and error
100	500	100	−100 Lose and error
		500	+500 Gain and correct
100	500	100	+100 Gain and error
		500	+500 Gain and correct
100	500	100	−100 Lose and correct
		500	−500 Lose and error

**Table 2 brainsci-13-01235-t002:** Behavioral characteristics (M ± SD).

	Risk-Seeker	Risk-Averse	*p* Value
Gender (M/F)	5/9	4/10	
Age	28.7 ± 7.6	28.5 ± 6.7	0.917
BART (adjusted value)	67.35 ± 10.4	14.55 ± 4.47	0.001
Gambling task performance:			
% Low-risk choice (100)	36.56 ± 10.4	45.74 ± 12.2	0.041
% High-risk choice (500)	60.29 ± 10.1	54.12 ± 4.6	0.049
Reaction time for 100	703.6 ± 116.8	702.2 ± 156.6	0.978
Reaction time for 500	660.9 ± 120.9	629 ± 121.10	0.491
Barratt impulsiveness scale			
Attentional impulsiveness	20.72 ± 3.25	22.07 ± 3.2	0.211
Motor impulsiveness	18.07 ± 3.6	19.92 ± 3.9	0.603
Non-planing impulsiveness	21.57 ± 4.12	22.35 ± 3.75	0.400
Total Barratt score	22.64 ± 4.78	24 ± 3.5	0.285

## Data Availability

Data available on request due to restrictions, e.g., privacy or ethics.

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
