# Peer review of "Neural Differences in Relation to Risk Preferences during Reward Processing: An Event-Related Potential Study"

_brainsci, 2023, doi:10.3390/brainsci13091235_

Round 1

Reviewer 1 Report

Neural differences in relation to risk preferences during reward processing: An Event-related potential study brainsci-2536683

This manuscript aimed to explore the neural correlates in relation to risk preferences during reward processing in non-clinical samples by using ERP. The results revealed that Low-risk option elicited relatively blunted stimulus-preceding negativity (SPN) response in risk seekers compared to risk averse, during the anticipation stage. Feedback-related negativity (FRN), during the outcome stage, was enhanced in response to greater losses in risk seekers but not in risk averse. Overall, this topic is interesting. However, some concerns appeared after reading the whole manuscript.

1. “Previous studies mainly examined risky behaviors during reward processing in clinical populations” this statement might be not appropriate since many of these investigations would use non-clinical sample as health control group. Thus, the evidence on non-clinical population might not be so limited as the authors stated. In addition, some important papers were missing in the current manuscript, which need to be reviewed and discussed, such as,

Deng, L., Li, Q., Zhang, M., Shi, P., & Zheng, Y. (2023). Distinct neural dynamics underlying risk and ambiguity during valued-based decision making. Psychophysiology, 60, e14201. https://doi.org/10.1111/psyp.14201

Teti Mayer, J.; Compagne, C.; Nicolier, M.; Grandperrin, Y.; Chabin, T.; Giustiniani, J.; Haffen, E.; Bennabi, D.; Gabriel, D. Towards a Functional Neuromarker of Impulsivity: Feedback-Related Brain Potential during Risky Decision-Making Associated with Self-Reported Impulsivity in a Non-Clinical Sample. Brain Sci. 2021, 11, 671. https://doi.org/10.3390/brainsci11060671

Thus, the research gaps and the novelties of current investigation should be revised.

2. Many other components also are involved in the risk preferences and reward processing, such as P300, then why did you not analyze those components

3. There are some female participants in the current sample, and the menstruation cycle would affect risk preferences and reward processing. Did you control the potential effects of menstruation cycle in female participants.

4. Some personal characteristics, such as impulsivity, would also influence risk preferences and reward processing, and did you measure these characteristics and treat them as covariates when analyzing the data?

5. How did you determine the sample size? Did you calculate the sample size needed before formal study? The current sample size seems too little to get reliable results.

References:

Lakens, D. (2022). Sample size justification. Collabra: Psychology, 8(1), 33267.

Larson, M. J., & Carbine, K. A. (2017). Sample size calculations in human electrophysiology (EEG and ERP) studies: A systematic review and recommendations for increased rigor. International Journal of Psychophysiology, 111, 33-41.

Clayson, P. E., Carbine, K. A., Baldwin, S. A., & Larson, M. J. (2019). Methodological reporting behavior, sample sizes, and statistical power in studies of eventrelated potentials: Barriers to reproducibility and replicability. Psychophysiology, 56(11), e13437.

6. For discussion part, the discussion about the potential reasons for discrepancies between the present findings and previous findings should be provided (line 384-387).

7. Future studies are desirable to consider multicomponent analysis to generate more de- 396
tailed results.
does not mean anything.

8. The limitation part needs to be added.

9. The behavioral result in the abstract part is not consistent with findings in the results part.

10. I recommend that the paper be thoroughly proofread and edited for languages and grammars, to enhance readership.

Moderate editing of English language required.

Author Response

Dear reviewer

We thank you for your appreciation of our work and for your insightful constructive comments. We have been able to incorporate changes to reflect most of the suggestions provided by the reviewers. We have highlighted all changes made within the manuscript. Here is a point-by-point response to the reviewer’s comments and concerns.

Reviewer 1, Comment 1: Previous studies mainly examined risky behaviors during reward processing in clinical populations” This statement might be not appropriate since many of these investigations would use non-clinical sample as health control group. Thus, the evidence on non-clinical populations might not be so limited as the authors stated. In addition, some important papers were missing in the current manuscript, which needs to be reviewed and discussed…

Authors’ Response: We agree with this comment. Consequently, we have added the suggested articles to the manuscript and toned down some statements regarding the limited literature on non-clinical populations. All changes have been highlighted in the introduction part.

Reviewer 1, Comment 2: Many other components also are involved in the risk preferences and reward processing, such as P300, then why did you not analyze those components?

Authors’ Response: we have already published an article extracted from this project concentrating on P300, but in Persian. We wanted to focus As it is explained in the manuscript lines 108-111.  

Reviewer 1, Comment 3: There are some female participants in the current sample, and the menstruation cycle would affect risk preferences and reward processing. Did you control the potential effects of the menstruation cycle in female participants?

Authors’ Response: Thank you for your suggestion, as you have raised an important and valuable point here. It would have been interesting to explore this aspect. However, in the case of our study, it seems slightly out of our scope. In fact, there would be several interesting variables that might potentially influence the results and are not typically measured, besides from menstruation cycle (e.g., socio-economic status). We chose to insert this as a possible limitation and as a suggestion for future studies. To our knowledge, no study has mentioned this point besides measuring risk propensities.

Reviewer 1, Comment 4: Some personal characteristics, such as impulsivity, would also influence risk preferences and reward processing, and did you measure these characteristics and treat them as covariates when analyzing the data?

Authors’ Response: we had actually used Barratt impulsiveness scale as a supplementary scale. We originally did not focus on this variable. However, it is now added to the manuscript based on your suggestion.

Reviewer 1, Comment 5: How did you determine the sample size? Did you calculate the sample size needed before formal study? The current sample size seems too little to get reliable results.

Authors’ Response: Thank you for your consideration. We agree that the final sample size of 28 might not be big enough to obtain a large effect size. However, some related studies in the literature used smaller sample sizes such as Zheng et al (2020) with a sample size of 24. Nonetheless, we incorporated the issue concerning sample size as a limitation of our study, in the light of your comment.

Reviewer 1, Comment 6: For the discussion part, the discussion about the potential reasons for discrepancies between the present findings and previous findings should be provided (line 384-387).

Authors’ Response: Thanks for your helpful comment. This part of the discussion has been changed and revised accordingly.

Reviewer 1, Comment 7: “Future studies are desirable to consider multicomponent analysis to generate more detailed results.” does not mean anything.

Authors’ Response: This sentence has been removed.

Reviewer 1, Comments 8: The limitation part needs to be added.

Authors’ Response: Agreed. The limitation section has been added based on your suggestions.

Reviewer 1, Comment 9: The behavioral result in the abstract part is not consistent with findings in the results part.

Authors’ Response: Thank you for pointing this out. The abstract has been changed consistently with the behavioral results.

Reviewer 1, Comment 10: I recommend that the paper be thoroughly proofread and edited for languages and grammar, to enhance readership.

Authors’ Response: we have incorporated your suggestion throughout the manuscript.

Reviewer 2 Report

Naghel and colleagues studied neural correlates of risk taking behaviours during reward processing. The study was well written and the results are quite clear. I however have a few comments.

Major comments:

1.    Could the authors please explain briefly why the short baseline of only 200 ms?. Did they anticipate artefacts with data close to the stimulus onset?, Or how far in time do we expect to see the SPN. There is no reasoning provided for this baseline selection.

2.    Is there any functional relevance of the earlier 50-200 ms before the FRN, especially the N100 peaks?. There seems to be a higher negative peak for the ‘high lose 500’ condition.

3.    It is not clear if there was averaging of electrodes to quantify activity for SPN and FRN. Description of Fig. 5 showed the plots presented were only activity at Fz. The authors showed earlier that they expected activity at Fz, C2, and Cz electrodes. Is there any reason analyses was done only at one electrode, and not a region of interest (ROI)?. This might be relevant because anatomically, ACC or any other brain region does not map physiologically to a single EEG electrode.  

Minor comments:

1.    Any reason why the group sizes where not balanced? 16 vs 15.

2.    Lines 169-170: The color does not match the table. Red means lose, and green means win. Sentence should rather be written as: ‘They were similar to targets in shape and size, but their colors changed to green or red, indicating a win or a loss, respectively’.

3.    Please label the axis of the plots (measurement units such as time in ‘seconds’, and waveform amplitudes such as ‘microvolts’ are missing as well)  – both the time series and the 2D topographic head plots, as well as the color bar to the 2D plot have no units.

4.    Simplify the 2D head plots to show clearly the decrease or increase in activity. You could remove the text. We cannot see the electrodes anyway, if the text is showing the electrode position, it is not visible enough.

Author Response

Dear reviewer

Thank you for giving us the opportunity to submit a revised draft of our manuscript. We appreciate the time and effort you and the reviewers have dedicated to providing valuable feedback on our manuscript. We have been able to incorporate changes to reflect most of the suggestions provided by the reviewers. Changes are highlighted within the manuscript.

Major comments

Reviewer 2, Comment 1: Could the authors please explain briefly why the short baseline of only 200 ms?. Did they anticipate artefacts with data close to the stimulus onset?, Or how far in time do we expect to see the SPN. There is no reasoning provided for this baseline selection.

Authors’ Response: Thank you for pointing this out. The baseline and epochs were chosen according to the literature and related articles using these components such as Zheng & Liu (2015) and Jung et al. (2001). Accordingly, relevant references have been added to this part of the manuscript and reported below for the reviewers’ perusal.

Jung, T.P., Makeig, S., Westerfield, M., Townsend, J., Courchesne, E., Sejnowski, T.J., 2001. Analysis and visualization of single-trial event-related potentials. Hum. Brain Mapp. 14 (3), 166–185.

Zheng, Y., & Liu, X. (2015). Blunted neural responses to monetary risk in high sensation seekers. Neuropsychologia, 71, 173-180.

Reviewer 2, Comment 2:  Is there any functional relevance of the earlier 50-200 ms before the FRN, especially the N100 peaks?. There seems to be a higher negative peak for the ‘high lose 500’ condition.

Authors’ Response: Thank you for pointing this. We have also noticed it. It might be the or error-related negativity (ERN). Both are important components but they rarely described at literature of ERP correlates of rewards (e.g., Glazer et al., 2018 as a review study). ERN is correlated with Error detection. It would have been interesting to explore this aspect. However, in the case of our study, it seems slightly out of scope. No error stimulus was used.

Reviewer 2, Comment 3: It is not clear if there was averaging of electrodes to quantify activity for SPN and FRN. Description of Fig. 5 showed the plots presented were only activity at Fz. The authors showed earlier that they expected activity at Fz, C2, and Cz electrodes. Is there any reason analyses was done only at one electrode, and not a region of interest (ROI)?. This might be relevant because anatomically, ACC or any other brain region does not map physiologically to a single EEG electrode.  

Authors’ Response: we agree. The analysis was also performed for C2 and Cz, But the difference between groups was significant just on the Fz electrode. Data on one significant electrode has been reported as previous studies do; such as Teti Mayer et al (2021) and Zheng et al (2020).

Minor comments

Reviewer 2, Comment 1: Any reason why the group sizes were not balanced? 16 vs 15.

Authors’ Response: Subjects were selected according to their score in BART task during a screening phase; therefore, the samples were unbalanced at the beginning. And finally, there were 14 subjects per group, since 3 subjects were removed (2 in one group and 1 in the other group) because of artifact detection in their EEG recordings.

Reviewer 2, Comment 2.  Lines 169-170: The color does not match the table. Red means lose, and green means win. Sentence should rather be written as: ‘They were similar to targets in shape and size, but their colors changed to green or red, indicating a win or a loss, respectively’.

Authors’ Response: Thanks a lot, this was fixed.

Reviewer 2, Comment 3. Please label the axis of the plots (measurement units such as time in ‘seconds’, and waveform amplitudes such as ‘microvolts’ are missing as well)  – both the time series and the 2D topographic head plots, as well as the color bar to the 2D plot have no units.

Authors’ Response: ok, thanks. This has been changed, accordingly.

Reviewer 2, Comment 4. Simplify the 2D head plots to show clearly the decrease or increase in activity. You could remove the text. We cannot see the electrodes anyway, if the text is showing the electrode position, it is not visible enough.

Authors’ Response: thank you, your suggestion has been implemented in plots by showing decrease and increase in activity more clearly.

Round 2

Reviewer 1 Report

Thanks for the revisions and some concerns remain.

1. As for the P300, the published Persian paper needs to be mentioned in the methodology part to help the readers get better understanding about how you use the data from this investigation.

2. The language seems not be improved much and still needs improvement.

The language seems not be improved much and still needs improvement.

Author Response

Dear reviewer

We are very much thankful for your deep and thorough review. The Persian article has been mentioned as you have suggested in comment 1. Regarding your second comment, the manuscript has been edited and proofread by a native expert. Most of the changes have been highlighted throughout the manuscript. The certificate from the agency is attached to the letter.

We truly appreciate your consideration.

Yours sincerely

Authors team

Reviewer 2 Report

I appreciate the efforts of the authors in answering all of my questions. I accept the reviewed manuscript in this current form. 

I hope a final edit is done to remove comments from the authors. 

Author Response

Dear reviewer
We greatly appreciate your assistance and valuable comments on our manuscript. We feel that it is much improved as a result of the changes we have made based on your comments. It is truly appreciated.

Yours sincerely
Authors team
